# Learning Graphical State Transitions

**Daniel D. Johnson**
Department of Computer Science
Harvey Mudd College
301 Platt Boulevard
ddjohnson@hmc.edu

## Abstract

Graph-structured data is important in modeling relationships between multiple entities, and can be used to represent states of the world as well as many data structures. Li et al. (2016) describe a model known as a Gated Graph Sequence Neural Network (GGS-NN) that produces sequences from graph-structured input. In this work I introduce the Gated Graph Transformer Neural Network (GGT-NN), an extension of GGS-NNs that uses graph-structured data as an intermediate representation. The model can learn to construct and modify graphs in sophisticated ways based on textual input, and also to use the graphs to produce a variety of outputs. For example, the model successfully learns to solve almost all of the bAbI tasks (Weston et al., 2016), and also discovers the rules governing graphical formulations of a simple cellular automaton and a family of Turing machines.

## 1 Introduction

Many different types of data can be formulated using a graph structure. One form of data that lends itself to a graphical representation is data involving relationships (edges) between entities (nodes). Abstract maps of places and paths between them also have a natural graph representation, where places are nodes and paths are edges. In addition, many data structures can be expressed in graphical form, including linked lists and binary trees.

Substantial research has been done on producing output when given graph-structured input (Kashima et al., 2003; Shervashidze et al., 2011; Perozzi et al., 2014; Bruna et al., 2013; Duvenaud et al., 2015). Of particular relevance to this work are Graph Neural Networks (Gori et al., 2005; Scarselli et al., 2009), or GNNs, which extend recursive neural networks by assigning states to each node in a graph based on the states of adjacent nodes. Recently Li et al. (2016) have modified GNNs to use gated state updates and to produce output sequences. The resulting networks, called GG-NNs and GGS-NNs, are successful at solving a variety of tasks with graph-structured input.

The current work further builds upon GG-NNs and GGS-NNs by allowing graph-structured intermediate representations, as well as graph-structured outputs. This is accomplished using a more flexible graph definition, along with a set of graph transformations which take a graph and other information as input and produce a modified version of the graph. This work also introduces the Gated Graph Transformer Neural Network model (GGT-NN), which combines these transformations with a recurrent input model to incrementally construct a graph given natural language input, and can either produce a final graph representing its current state, or use the graph to produce a natural language output.

Extending GG-NNs in this way opens up a wide variety of applications. Since many types of data can be naturally expressed as a graph, it is possible to train a GGT-NN model to manipulate a meaningful graphical internal state. In this paper I demonstrate the GGT-NN model on the bAbI task dataset, which contains a set of stories about the state of the world. By encoding this state as a graph and providing these graphs to the model at training time, a GGT-NN model can be trained to construct the correct graph from the input sentences and then answer questions based on this internal graph. I also demonstrate that this architecture can learn complex update rules by training it to model a simple 1D cellular automaton and arbitrary 4-state Turing machines. This requires the network to learn how to transform its internal state based on the rules of each task.

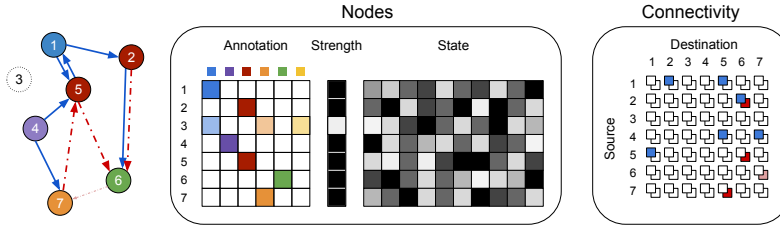

Figure 1: Diagram of the differentiable encoding of a graphical structure, as described in section 3. On the left, the desired graph we wish to represent, in which there are 6 node types (shown as blue, purple, red, orange, green, and yellow) and two edge types (shown as blue/solid and red/dashed). Node 3 and the edge between nodes 6 and 7 have a low strength. On the right, depictions of the node and edge matrices: annotations, strengths, state, and connectivity correspond to $\mathbf{x}_v$, $s_v$, $\mathbf{h}_v$, and $\mathcal{C}$, respectively. Saturation represents the value in each cell, where white represents 0, and fully saturated represents 1. Note that each node's annotation only has a single nonzero entry, corresponding to each node having a single well-defined type, with the exception of node 3, which has an annotation that does not correspond to a single type. State vectors are shaded arbitrarily to indicate that they can store network-determined data. The edge connectivity matrix $\mathcal{C}$ is three dimensional, indicated by stacking the blue-edge cell on top of the red-edge cell for a given source-destination pair. Also notice the low strength for cell 3 in the strength vector and for the edge between node 6 and node 7 in the connectivity matrix.

## 2 BACKGROUND

### 2.1 GRU

Gated Recurrent Units (GRU) are a type of recurrent network cell introduced by Cho et al. (2014). Each unit uses a reset gate $r$ and an update gate $z$, and updates according to

$$\mathbf{r}^{(t)} = \sigma\left(\mathbf{W}_r\mathbf{x}^{(t)} + \mathbf{U}_r\mathbf{h}^{(\mathbf{t-1})} + \mathbf{b}_r\right) \qquad \mathbf{z}^{(t)} = \sigma\left(\mathbf{W}_z\mathbf{x}^{(t)} + \mathbf{U}_z\mathbf{h}^{(\mathbf{t-1})} + \mathbf{b}_z\right)$$

$$\widetilde{\mathbf{h}}^{(t)} = \phi\left(\mathbf{W}\mathbf{x} + \mathbf{U}\left(\mathbf{r}^{(t)} \odot \mathbf{h}^{(\mathbf{t-1})}\right) + \mathbf{b}\right) \qquad \mathbf{h}^{(t)} = \mathbf{z} \odot \mathbf{h}^{(t-1)} + (1 - \mathbf{z}) \odot \widetilde{\mathbf{h}}^{(t)}$$

where $\sigma$ is the logistic sigmoid function, $\phi$ is an activation function (here $\tanh$ is used), $\mathbf{x}^{(t)}$ is the input vector at timestep $t$, $\mathbf{h}^{(t)}$ is the hidden output vector at timestep $t$, and $\mathbf{W}, \mathbf{U}, \mathbf{W}_r, \mathbf{U}_r, \mathbf{W}_z$, $\mathbf{U}_z, \mathbf{b}, \mathbf{b}_r$ and $\mathbf{b}_z$ are learned weights and biases. Note that $\odot$ denotes elementwise multiplication.

### 2.2 GG-NN AND GGS-NN

The Gated Graph Neural Network (GG-NN) is a form of graphical neural network model described by Li et al. (2016). In a GG-NN, a graph $\mathcal{G} = (\mathcal{V}, \mathcal{E})$ consists of a set $V$ of nodes $v$ with unique values and a set $\mathcal{E}$ of directed edges $e = (v, v') \in \mathcal{V} \times \mathcal{V}$ oriented from $v$ to $v'$. Each node has an annotation $\mathbf{x}_v \in \mathbb{R}^N$ and a hidden state $\mathbf{h}_v \in \mathbb{R}^D$, and each edge has a type $y_e \in \{1, \cdots, M\}$.

GG-NNs operate by first initializing the state $\mathbf{h}_v$ of each node to correspond to the annotation $\mathbf{x}_v$. Then, a series of propagation steps occur. In each step, information is transferred between nodes across the edges, and the types of edge determine what information is sent. Each node sums the input it receives from all adjacent nodes, and uses that to update its own internal state, in the same manner as a GRU cell. Finally, the states of all nodes are used either to create a graph-level aggregate output, or to classify each individual node.

GGS-NNs extend GG-NNs by performing a large number of propagation-output cycles. At each stage, two versions of the GG-NN propagation process are run. The first is used to predict an output for that timestep, and the second is used to update the annotations of the nodes for the next timestep. This allows GGS-NNs to predict a sequence of outputs from a single graph.

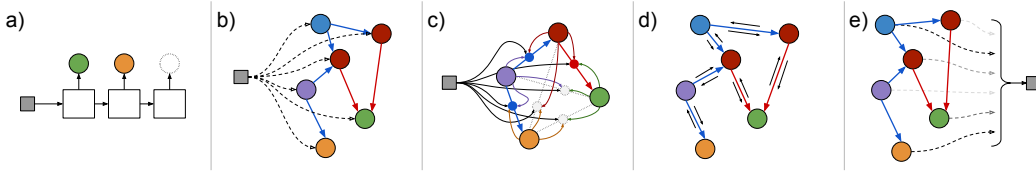

Figure 2: Summary of the graph transformations. Input and output are represented as gray squares. a) Node addition ($\mathcal{T}_{\text{add}}$), where the input is used by a recurrent network (white box) to produce new nodes, of varying annotations and strengths. b) Node state update ($\mathcal{T}_{\mathbf{h}}$), where each node receives input (dashed line) and updates its internal state. c) Edge update ($\mathcal{T}_{\mathcal{C}}$), where each existing edge (colored) and potential edge (dashed) is added or removed according to the input and states of the adjacent nodes (depicted as solid arrows meeting at circles on each edge). d) Propagation ($\mathcal{T}_{\text{prop}}$), where nodes exchange information along the current edges, and update their states. e) Aggregation ($\mathcal{T}_{\text{repr}}$), where a single representation is created using an attention mechanism, by summing information from all nodes weighted by relevance (with weights shown by saturation of arrows).

## 3 DIFFERENTIABLE GRAPH TRANSFORMATIONS

In this section, I describe some modifications to the graph structure to make it fully differentiable, and then propose a set of transformations which can be applied to a graph structure in order to transform it. In particular, I redefine a graph $\mathcal{G} = (\mathcal{V}, \mathcal{C}) \in \Gamma$ as a set $V$ of nodes $v$, and a *connectivity matrix* $\mathcal{C} \in \mathbb{R}^{|\mathcal{V}| \times |\mathcal{V}| \times Y}$, where $Y$ is the number of possible edge types. As before, each node has an annotation $\mathbf{x}_v \in \mathbb{R}^N$ and a hidden state $\mathbf{h}_v \in \mathbb{R}^D$. However, there is an additional constraint that $\sum_{j=1}^{N} x_{v,j} = 1$. One can then interpret $x_{v,j}$ as the level of belief that $v$ should have type $j$ out of $N$ possible *node types*. Each node also has a strength $s_v \in [0, 1]$. This represents the level of belief that node $v$ should exist, where $s_v = 1$ means the node exists, and $s_v = 0$ indicates that the node should not exist and thus should be ignored.

Similarly, elements of $\mathcal{C}$ are constrained to the range $[0, 1]$, and thus one can interpret $\mathcal{C}_{v,v',y}$ as the level of belief that there should be a directed edge of type $y$ from $v$ to $v'$. (Note that it is possible for there to be edges of multiple types between the same two nodes $v$ and $v'$, i.e. it is possible for $\mathcal{C}_{v,v',y} = \mathcal{C}_{v,v',y'} = 1$ where $y \neq y'$.) Figure 1 shows the values of $\mathbf{x}_v$, $s_v$, $\mathbf{h}_v$, and $\mathcal{C}$ corresponding to a particular graphical structure.

There are five classes of graph transformation:

- a) Node addition ($\mathcal{T}_{\text{add}}$), which modifies a graph by adding new nodes and assigning them annotations $\mathbf{x}_v$ and strengths $s_v$ based on an input vector.
- b) Node state update ($\mathcal{T}_{\mathbf{h}}$), which modifies the internal state of each node using an input vector (similar to a GRU update step). Optionally, different input can be given to nodes of each type, based on direct textual references to specific node types. This version is called a direct reference update ($\mathcal{T}_{\mathbf{h},\text{direct}}$).
- c) Edge update ($\mathcal{T}_{\mathcal{C}}$), which modifies the edges between each pair of nodes based on the internal states of the two nodes and an external input vector.
- d) Propagation ($\mathcal{T}_{\text{prop}}$), which allows nodes to trade information across the existing edges and then update their internal states based on the information received.
- e) Aggregation ($\mathcal{T}_{\text{repr}}$), which uses an attention mechanism to select relevant nodes and then generates a graph-level output.

Each transformation has its own trainable parameters. Together, these transformations can be combined to process a graph in complex ways. An overview of these operations is shown in Figure 2. For details about the implementation of each of these transformations, see Appendix B.

## 4 GATED GRAPH TRANSFORMER NEURAL NETWORK (GGT-NN)

In this section I introduce the Gated Graph Transformer Neural Network (GGT-NN), which is constructed by combining a series of these transformations. Depending on the configuration of the transformations, a GGT-NN can take textual or graph-structured input, and produce textual or graph-

---

**Algorithm 1** Graph Transformation Pseudocode

---

1: $\mathcal{G} \leftarrow \varnothing$ 11: $\mathcal{G} \leftarrow \mathcal{T}_{\mathrm{add}}(\mathcal{G}, [\mathbf{i}^{(k)} \ \mathbf{h}_{\mathcal{G}}^{\mathrm{add}}])$

2: **for** $k$ from 1 to $K$ **do** 12: $\mathcal{G} \leftarrow \mathcal{T}_{\mathcal{C}}(\mathcal{G}, \mathbf{i}^{(k)})$

3: $\mathcal{G} \leftarrow \mathcal{T}_{\mathbf{h}}(\mathcal{G}, \mathbf{i}^{(k)})$ 13: **end for**

4: **if** direct reference enabled **then** 14: $\mathcal{G} \leftarrow \mathcal{T}_{\mathbf{h}}^{\mathrm{query}}(\mathcal{G}, \mathbf{i}^{\mathrm{query}})$

5: $\mathcal{G} \leftarrow \mathcal{T}_{\mathbf{h},\mathrm{direct}}(\mathcal{G}, \mathbf{D}^{(k)})$ 15: **if** direct reference enabled **then**

6: **end if** 16: $\mathcal{G} \leftarrow \mathcal{T}_{\mathbf{h},\mathrm{direct}}^{\mathrm{query}}(\mathcal{G}, \mathbf{D}^{\mathrm{query}})$

7: **if** intermediate propagation enabled **then** 17: **end if**

8: $\mathcal{G} \leftarrow \mathcal{T}_{\mathrm{prop}}(\mathcal{G})$ 18: $\mathcal{G} \leftarrow \mathcal{T}_{\mathrm{prop}}^{\mathrm{query}}(\mathcal{G})$

9: **end if** 19: $\mathbf{h}_{\mathcal{G}}^{\mathrm{answer}} \leftarrow \mathcal{T}_{\mathrm{repr}}^{\mathrm{query}}(\mathcal{G})$

10: $\mathbf{h}_{\mathcal{G}}^{\mathrm{add}} \leftarrow \mathcal{T}_{\mathrm{repr}}(\mathcal{G})$ 20: **return** $f_{\mathrm{output}}(\mathbf{h}_{\mathcal{G}}^{\mathrm{answer}})$

---

structured output. Here I describe one particular GGT-NN configuration, designed to build and modify a graph based on a sequence of input sentences, and then produce an answer to a query.

When run, the model performs the following: For each sentence $k$, each word is converted to a one-hot vector $\mathbf{w}_l^{(k)}$, and the sequence of words (of length $L$) is passed through a GRU layer to produce a sequence of partial-sentence representation vectors $\mathbf{p}_l^{(k)}$. The full sentence representation vector $\mathbf{i}^{(k)}$ is initialized to the last partial representation vector $\mathbf{p}_L^{(k)}$. Furthermore, a direct-reference input matrix $\mathbf{D}^{(k)}$ is set to the sum of partial representation vectors corresponding to the words that directly reference a node type, i.e. $\mathbf{D}_n^{(k)} = \sum_{l \in R_n} \mathbf{p}_l^{(k)}$ where $R_n$ is the set of words in sentence $k$ that directly refer to node type $n$. This acts like an attention mechanism, by accumulating the partial representation vectors for the words that directly reference each type, and masking out the vectors corresponding to other words.

Next, a series of graph transformations are applied, as depicted in Algorithm 1. Depending on the task, direct reference updates and per-sentence propagation can be enabled or disabled. The output function $f_{\mathrm{output}}$ will depend on the specific type of answer desired. If the answer is a single word, $f_{\mathrm{output}}$ can be a multilayer perceptron followed by a softmax operation. If the answer is a sequence of words, $f_{\mathrm{output}}$ can use a recurrent network (such as a GRU) to produce a sequence of outputs. Note that transformations with different superscripts ($\mathcal{T}_{\mathbf{h}}$ and $\mathcal{T}_{\mathbf{h}}^{\mathrm{query}}$, for instance) refer to similar transformations with different learned weights.

Since the processing of the input and all of the graph transformations are differentiable, at this point the network output can be compared with the correct output for that query and used to update the network parameters, including both the GRU parameters used when processing the input and the internal weights associated with each transformation.

## 4.1 SUPERVISION

As with many supervised models, one can evaluate the loss based on the likelihood of producing an incorrect answer, and then minimize the loss by backpropagation. However, based on initial experiments, the model appeared to require additional supervision to extract meaningful graph-structured data. To provide this additional supervision, I found it beneficial to provide the correct graph at each timestep and train the network to produce that graph. This occurs in two stages, first when new nodes are proposed, and then when edges are adjusted. For the edge adjustment, the edge loss between a correct edge matrix $\mathcal{C}^*$ and the computed edge matrix $\mathcal{C}$ is given by

$$\mathcal{L}_{\mathrm{edge}} = -\mathcal{C}^* \cdot \ln(\mathcal{C}) - (1 - \mathcal{C}^*) \cdot \ln(1 - \mathcal{C}).$$

The node adjustment is slightly more complex. Multiple nodes are added in each timestep, but the order of those nodes is arbitrary, and only their existence is important. Thus it should be possible for the network to determine the optimal ordering of the nodes. In fact, this is important because there is no guarantee that the nodes will be ordered consistently in the training data.

Vinyals et al. (2016) demonstrate a simple method for training a network to output unordered sets: the network produces a sequence of outputs, and these outputs are compared with the closest order-

ing of the training data, i.e., the ordering of the training data which would produce the smallest loss when compared with the network output. Vinyals et al. show that when using this method, the network arbitrarily chooses an ordering which may not be the optimal ordering for the task. However, in this case any ordering should be sufficient, and I found the arbitrary orderings selected in this way to work well in practice. In particular, letting $s^*_{\pi(v)}$ and $\mathbf{x}^*_{\pi(v)}$ denote the correct strength and annotations of node $v$ under ordering $\pi$, the loss becomes

$$\mathcal{L}_{\text{node}} = -\max_\pi \sum_{v=|\mathcal{V}_{\text{old}}|+1}^{|\mathcal{V}_{\text{new}}|} s^*_{\pi(v)} \ln(s_v) + (1 - s^*_{\pi(v)}) \ln(1 - s_v) + \mathbf{x}^*_{\pi(v)} \cdot \ln(\mathbf{x}_v).$$

At this point the correct values $\mathcal{C}^*$, $s^*_{\pi(v)}$ and $\mathbf{x}^*_{\pi(v)}$ are substituted into the graph for further processing. Note that only the edges and the new nodes are replaced by the supervision. The hidden states of all existing nodes are propagated without adjustment.

## 4.2 Other Transformation Configurations

The structure described in Algorithm 1 is designed for question-answering tasks. However, due to the composability of the individual graph transformations, other configurations could be used to solve other tasks that operate on structured data.

For instance, if a task consists of tracking relationships between a fixed set of objects, one could construct a version of the model that does not use the new-nodes transformation ($\mathcal{T}_{\text{add}}$), but instead only modifies edges. If the task was to extract information from an existing graph, a structure similar to the GGS-NNs could be built by using only the propagation and aggregation transformations. If the task was to construct a graph based on textual input, the query processing steps could be omitted, and instead the final graph could be returned for processing. And if information should be gathered from a sequence of graphs instead of from a single graph, the query processing steps could be modified to run in parallel on the full sequence of graphs and extract information from each graph. This last modification is demonstrated in Appendix D.

## 5 Experiments

### 5.1 bAbI Tasks

I evaluated the GGT-NN model on the bAbI tasks, a set of simple natural-language tasks, where each task is structured as a sequence of sentences followed by a query (Weston et al., 2016). The generation procedure for the bAbI tasks includes a "Knowledge" object for each sentence, representing the current state of knowledge after that sentence. I exposed this knowledge object in graph format, and used this to train a GGT-NN in supervised mode. The knowledge object provides names for each node type, and direct reference was performed based on these names: if a word in the sentence matched a node type name, it was parsed as a direct reference to all nodes of that type. For details on this graphical format, see Appendix C.

### 5.1.1 Analysis and Results

I trained two versions of the GGT-NN model for each task: one with and one without direct reference. Tasks 3 and 5, which involve a complex temporal component, were trained with intermediate propagation, whereas all of the other tasks were not because the structure of the tasks made such complexity unnecessary. Most task models were configured to output a single word, but task 19 (pathfinding) used a GRU to output multiple words, and task 8 (listing) was configured to output a strength for each possible word to allow multiple words to be selected without having to consider ordering.

Results are shown in Tables 1 and 2. The GGT-NN model was able to reach 95% accuracy in all but one of the tasks, and reached 100% accuracy in eleven of them (see Table 2). Additionally, for fourteen of the tasks, the model was able to reach 95% accuracy using 500 or fewer of the 1000 training examples (see Table 1).

The only task that the GGT-NN was unable to solve with 95% accuracy was task 17 (Positional Reasoning), for which the model was not able to attain a high accuracy. Task 17 has a larger number

| Task | GGT-NN + direct ref | GGT-NN | Task | GGT-NN + direct ref | GGT-NN |
|---|---|---|---|---|---|
| 1 - Single Supporting Fact | 100 | 1000 | 11 - Basic Coreference | 100 | 1000 |
| 2 - Two Supporting Facts | 250 | - | 12 - Conjunction | 500 | 1000 |
| 3 - Three Supporting Facts | 1000 | - | 13 - Compound Coref. | 100 | 1000 |
| 4 - Two Arg. Relations | 1000 | 1000 | 14 - Time Reasoning | 1000 | - |
| 5 - Three Arg. Relations | 500 | - | 15 - Basic Deduction | 500 | 500 |
| 6 - Yes/No Questions | 100 | - | 16 - Basic Induction | 100 | 500 |
| 7 - Counting | 250 | - | 17 - Positional Reasoning | - | - |
| 8 - Lists/Sets | 250 | 1000 | 18 - Size Reasoning | 1000 | - |
| 9 - Simple Negation | 250 | - | 19 - Path Finding | 500 | - |
| 10 - Indefinite Knowledge | 1000 | - | 20 - Agent's Motivations | 250 | 250 |

Table 1: Number of training examples needed before the GGT-NN model could attain $\leq 5\%$ error on each of the bAbI tasks. Experiments were run with 50, 100, 250, 500, and 1000 examples. "GGT-NN + direct ref." denotes the performance of the model with direct reference, and "GGT-NN" denotes the performance of the model without direct reference. Dashes indicate that the model was unable to reach the desired accuracy with 1000 examples.

| | 1,000 examples | | | | | | 10,000 examples | | | | | |
|---|---|---|---|---|---|---|---|---|---|---|---|---|
| Task | GGT-NN + direct ref | GGT-NN | LSTM | MemNN | MemN2N | EntNet | NTM | D-NTM | MemN2N* | DNC | DMN+ | EntNet |
| 1 | **0** | **0.7** | 50.0 | **0** | **0** | **0.7** | 31.5 | **4.4** | **0** | **0** | **0** | **0** |
| 2 | **0** | 5.7 | 80.0 | **0** | 8.3 | 56.4 | 54.5 | 27.5 | **0.3** | **0.4** | **0.3** | **0.1** |
| 3 | **1.3** | 12.0 | 80.0 | **0** | 40.3 | 69.7 | 43.9 | 71.3 | **2.1** | **1.8** | **1.1** | **4.1** |
| 4 | **1.2** | **2.2** | 39.0 | **0** | **2.8** | **1.4** | **0** | **0** | **0** | **0** | **0** | **0** |
| 5 | **1.6** | 10.9 | 30.0 | **2.0** | 13.1 | **4.6** | **0.8** | **1.7** | **0.8** | **0.8** | **0.5** | **0.3** |
| 6 | **0** | 7.7 | 52.0 | **0** | 7.6 | 30.0 | 17.1 | **1.5** | **0.1** | **0** | **0** | **0.2** |
| 7 | **0** | 5.6 | 51.0 | 15.0 | 17.3 | 22.3 | 17.8 | 6.0 | **2.0** | **0.6** | **2.4** | **0** |
| 8 | **0** | **3.3** | 55.0 | 9.0 | 10.0 | 19.2 | 13.8 | **1.7** | **0.9** | **0.3** | **0** | **0.5** |
| 9 | **0** | 11.6 | 36.0 | **0** | 13.2 | 31.5 | 16.4 | **0.6** | **0.3** | **0.2** | **0** | **0.1** |
| 10 | **3.4** | 28.6 | 56.0 | **2.0** | 15.1 | 15.6 | 16.6 | 19.8 | **0** | **0.2** | **0** | **0.6** |
| 11 | **0** | **0.2** | 28.0 | **0** | **0.9** | 8.0 | 15.2 | **0** | **0** | **0** | **0** | **0.3** |
| 12 | **0.1** | **0.7** | 26.0 | **0** | **0.2** | **0.8** | 8.9 | 6.2 | **0** | **0** | **0.2** | **0** |
| 13 | **0** | **0.8** | 6.0 | **0** | **0.4** | 9.0 | 7.4 | 7.5 | **0** | **0** | **0** | **1.3** |
| 14 | **2.2** | 55.1 | 73.0 | **1.0** | **1.7** | 62.9 | 24.2 | 17.5 | **0.2** | **0.4** | **0.2** | **0** |
| 15 | **0.9** | **0** | 79.0 | **0** | **0** | 57.8 | 47.0 | **0** | **0** | **0** | **0** | **0** |
| 16 | **0** | **0** | 77.0 | **0** | **1.3** | 53.2 | 53.6 | 49.6 | 51.8 | 55.1 | 45.3 | **0.2** |
| 17 | 34.5 | 48.0 | 49.0 | 35.0 | 51.0 | 46.4 | 25.5 | **1.2** | 18.6 | 12.0 | **4.2** | **0.5** |
| 18 | **2.1** | 10.6 | 48.0 | **5.0** | 11.1 | 8.8 | **2.2** | **0.2** | 5.3 | **0.8** | **2.1** | **0.3** |
| 19 | **0** | 70.6 | 92.0 | 64.0 | 82.8 | 90.4 | **4.3** | 39.5 | **2.3** | **3.9** | **0** | **2.3** |
| 20 | **0** | **1.0** | 9.0 | **0** | **0** | **2.6** | **1.5** | **0** | **0** | **0** | **0** | **0** |

Table 2: Error rates of various models on the bAbI tasks. Bold indicates $\leq 5\%$ error. For descriptions of each of the tasks, see Table 1. "GGT-NN + direct ref." denotes the GGT-NN model with direct reference, and "GGT-NN" denotes the version without direct reference. See text for details regarding the models used for comparison. Results from LSTM and MemNN reproduced from Weston et al. (2016). Results from other existing models reproduced from Henaff et al. (2016).

of possible entities than the other tasks: each entity consists of a color (chosen from five options) and a shape (chosen from four shapes), for a total of 20 unique entities that must be represented separately. Additionally, the stories are much shorter than those in other tasks (2 facts for each set of 8 questions). It is likely that these additional complexities caused the network performance to suffer.

For comparison, accuracy on the bAbI tasks is also included for a simple sequence-to-sequence LSTM model and for a variety of existing state-of-the-art approaches (see Table 2): a simple sequence-to-sequence LSTM model, as implemented in Weston et al. (2016), a modified Memory Network model (MemNN, Weston et al., 2016), End-To-End Memory Network (MemN2N, Sukhbaatar et al., 2015), Recurrent Entity Network (EntNet, Henaff et al., 2016), Neural Turing Machine (NTM, Graves et al., 2014), Dynamic NTM (D-NTM, Gulcehre et al., 2016), a larger version of the MemN2N model with weight tying and nonlinearity (MemN2N*, Sukhbaatar et al., 2015), Differentiable Neural Computer (DNC, Graves et al., 2016), and Dynamic Memory Network (DMN+, Xiong et al., 2016). Although the GGT-NN model was trained using only 1,000 training examples, results using 10,000 examples have also been reproduced here for comparison. Also, it is important to note that the GGT-NN and MemNN models were trained with strong supervision: the GGT-NN model was trained with full graph information, and the MemNN model was trained with information on which sentences were relevant to the query. All other models were trained end-to-end without additional supervision.

Since the GGT-NN and MemNN models are both strongly supervised, it is interesting to note that each approach outperforms the other on a subset of the tasks. In particular, the GGT-NN model with direct reference attains a higher level of accuracy on the following tasks, with an improvement of 0.4-64% depending on the task: task 5 (0.4%), task 7 (15%), task 8 (9%), task 17 (0.5%), task 18 (2.9%), and task 19 (64%). This may indicate that a graphical representation is superior to a list of sentence memories for solving these tasks. On the other hand, the MemNN model outperforms the GGT-NN model (0.1-2.9% greater accuracy) on tasks 3, 4, 10, 12, 14, and 15.

Of particular interest is the performance on task 19, the pathfinding task, for which the GGT-NN model with direct reference performs better than all but one of the other models (DMN+), and shows a large improvement over the performance of the MemNN model. This is reasonable, since pathfinding is a task that is naturally suited to a graphical representation. The shortest path between two nodes can be easily found by sending information across all paths away from one of the nodes in a distributed fashion, which the GGT-NN model allows. Note that the preexisting GGS-NN model (discussed in Section 2.2) was also able to successfully learn the pathfinding task, but required the input to be preprocessed into graphical form even when evaluating the model, and thus could not be directly evaluated on the textual form of any of the bAbI tasks (Li et al., 2016). The current results demonstrate that the proposed GGT-NN model is able to solve the pathfinding task when given textual input.

Similarly, both variants of the GGT-NN model show improvement over many other models on task 16, the induction task. Solving the induction task requires being able to infer relationships based on similarities between entities. (One example from this task: *Lily is a swan. Lily is white. Bernhard is green. Greg is a swan. What color is Greg? A:white.*) In a graphical setting, this can be done by following a sequence of edges (Greg $\to$ swan $\to$ Lily $\to$ white), and the performance of the GGT-NN model indicates that this task is particularly suited to such a representation.

In general, the GGT-NN model with direct reference performs better than the model without it. The model with direct reference reaches 95% accuracy on 19/20 of the bAbI tasks, while the model without direct reference reaches that level of accuracy on 9/20 of the tasks (see Table 2). Additionally, when compared to the direct-reference model, the model without direct reference requires more training examples in order to reach the accuracy threshold (see Table 1). This indicates that, although the model can be used without direct reference, adding direct reference greatly improves the training of the model.

## 5.2 RULE DISCOVERY TASKS

To demonstrate the power of GGT-NN to model a wide variety of graph-based problems, I applied the GGT-NN to two additional tasks. In each task, a sequence of data structures were transformed into a graphical format, and the GGT-NN was tasked with predicting the data for the next timestep

|  | Original Task | Generalization: 20 | Generalization: 30 |
|---|---|---|---|
| **Automaton** | 100.0% | 87.0% | 69.5% |
| **Turing** | 99.9% | 90.4% | 80.4% |

Table 3: Accuracy of GGT-NN on the Rule 30 Automaton and Turing Machine tasks.

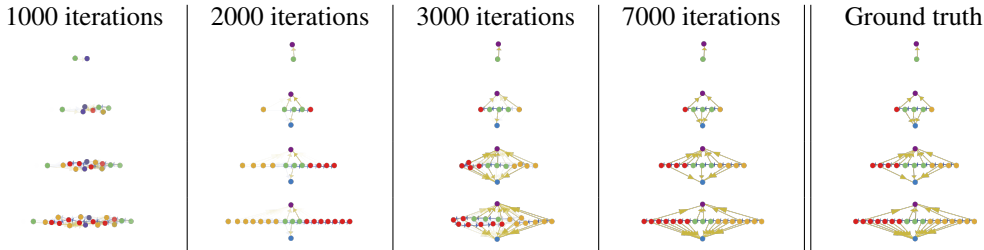

Figure 3: Visualization of network performance on the Rule 30 Automaton task. Top node (purple) represents zero, bottom node (blue) represents 1, and middle nodes (green, orange, and red) represent individual cells. Blue edges indicate adjacent cells, and gold edges indicate the value of each cell. Three timesteps occur between each row.

based on the current timestep. No additional information was provided as textual input; instead, the network was tasked with learning the rules governing the evolution of the graph structure over time.

### 5.2.1 CELLULAR AUTOMATON TASK

The first task used was a 1-dimensional cellular automaton, specifically the binary cellular automaton known as Rule 30 (Wolfram, 2002). Rule 30 acts on an infinite set of cells, each with a binary state (either 0 or 1). At each timestep, each cell deterministically changes state based on its previous state and the states of its neighbors. In particular, the update rules are

| Current neighborhood | 111 | 110 | 101 | 100 | 011 | 010 | 001 | 000 |
|---|---|---|---|---|---|---|---|---|
| Next value | **0** | **0** | **0** | **1** | **1** | **1** | **1** | **0** |

Cell states can be converted into graphical format by treating the cells as a linked list. Each of the cells is represented by a node with edges connecting it to the cell's neighbors, and a value edge is used to indicate whether the cell is 0 or 1. This format is described in more detail in Appendix C.

### 5.2.2 TURING MACHINES

The second task was simulating an arbitrary 2-symbol 4-state Turing machine. A Turing machine operates on an infinite tape of cells, each containing a symbol from a finite set of possible symbols. It has a head, which points at a particular cell and can read and write the symbol at that cell. It also has an internal state, from a finite set of states. At each timestep, based on the current state and the contents of the cell at the head, the machine writes a new symbol, changes the internal state, and can move the head left or right or leave it in place. The action of the machine depends on a finite set of rules, which specify the actions to take for each state-symbol combination. Note that the version of Turing machine used here has only 2 symbols, and requires that the initial contents of the tape be all 0 (the first symbol) except for finitely many 1s (the second symbol).

When converting a Turing machine to graphical format, the tape of the machine is modeled as a linked list of cells. Additionally, each state of the machine is denoted by a state node, and edges between these nodes encode the transition rules. There is also a head node, which connects both to the current cell and to the current state of the machine. See Appendix C for more details.

### 5.2.3 ANALYSIS AND RESULTS

The GGT-NN model was trained on 1000 examples of the Rule 30 automaton with different initial states, each of which simulated 7 timesteps of the automaton, and 20,000 examples of Turing

machines with different rules and initial tape contents, each of which simulated 6 timesteps of the Turing machine. Performance was then evaluated on 1000 new examples generated with the same format. The models were evaluated by picking the most likely graph generated by the model, and comparing it with the correct graph. The percent accuracy denotes the fraction of the examples for which these two graphs were identical at all timesteps. In addition to evaluating the performance on identical tasks, the generalization ability of the models was also assessed. The same trained models were evaluated on versions of the task with 20 and 30 timesteps of simulation.

Results are shown in Table 3. The models successfully learned the assigned tasks, reaching high levels of accuracy for both tasks. Additionally, the models show the ability to generalize to large inputs, giving a perfect output in the majority of extended tasks. For visualization purposes, Figure 3 shows the model at various stages of training when evaluated starting with a single 1 cell.

# 6  COMPARISON WITH RELATED WORK

Many methods have been proposed for combining neural networks with graphs. These methods generally require the input to the network to be in graphical format. For instance, GNNs and GGS-NNs take a graph as input, and propagate information between nodes according to the graph structure (Gori et al., 2005; Scarselli et al., 2009; Li et al., 2016). Similarly, graph convolutional networks extract information from an existing graph structure by using approximations to spectral graph convolutions (Kipf & Welling, 2016). These methods are similar to GGT-NNs in that they all store information in the nodes of a graph and use edges to determine how information flows. However, they all use a graph with fixed structure, and can only accept graphical data. The GGT-NN model, on the other hand, allows the graph structure to be built and modified based on unstructured input.

Giles et al. (1992) describe a method for extracting a finite state machine from a trained recurrent neural network by quantizing the hidden states of the network, recording all possible state transitions, and using them to construct a minimal directed graph representing the state machine. This method, however, requires postprocessing of the network to extract the graph, and is limited to extracting graphs that represent state machines. Additionally, although the FSM extraction method described by Giles et al. (1992) and the GGT-NN model both produce graphs using neural networks, the goals are different: the FSM extraction method aims to learn a single graph that can classify sequences, whereas the GGT-NN model aims to learn a neural network that can manipulate graphs.

The lifted relational neural network (LRNN) is another approach to working with structured data (Sourek et al., 2015). LRNNs require the input to be formatted as a combination of weighted predicate logic statements, encompassing both general rules and specific known facts. For each training example, the statements are used to construct a "ground neural network", with a connection pattern determined by the dependencies between the statements. LRNNs can learn to extract information by adjusting the weights of each statement, but require the rules to be composed by hand based on the task structure. Furthermore, unlike in GGT-NNs, a LRNN has no internal state associated with the objects it describes (which are instead represented by single neurons), and the relationships between objects cannot be constructed or modified by the network.

Multiple recent architectures have included differentiable internal states. Memory Networks, as described in Weston et al. (2014), and the fully differentiable end-to-end memory networks, described in Sukhbaatar et al. (2015), both utilize a differentiable long-term memory component, consisting of a set of memories that are produced by encoding the input sentences. To answer a query, an attention mechanism is used to select a subset of these memories, and the resulting memories are processed to produce the desired output. Differentiable Neural Computers (DNCs), described in Graves et al. (2016), interact with a fixed-size memory using a set of read and write "heads", which can be moved within the memory either by searching for particular content or by following temporal "links of association" that track the order in which data was written.

Memory networks and DNCs share with the GGT-NN model the ability to iteratively construct an internal state based on textual input, and use that internal state to answer questions about the underlying structured data. However, in these models, the structure of the internal state is implicit: although the network can store and work with structured data, the actual memory consists of a set of vectors that cannot be easily interpreted, except by monitoring the network access patterns. The GGT-NN model, on the other hand, explicitly models the internal state as a graph with labeled

nodes and edges. This allows the produced graph to be extracted, visualized, and potentially used in downstream applications that require graph-structured input.

Hierarchical Attentive Memory (HAM) is a memory-based architecture that consists of a binary tree built on top of an input sequence (Andrychowicz & Kurach, 2016). A recurrent controller accesses the HAM module by performing a top-down search through the tree, at each stage choosing to attend to either the left or right subtrees. Once this process reaches a leaf, the value of the leaf is provided to the controller to use in predicting the next output, and this leaf's value can be updated with a new value. This architecture is especially suited toward sequence-based tasks, and has been shown to generalize to longer sequences very efficiently due to the tree structure. However, it is unclear whether a HAM module would work well with non-sequential structured data, since the tree structure is fixed by the network.

One advantage of the GGT-NN model over existing works is that it can process data in a distributed fashion. Each node independently processes its surroundings, which can be beneficial for complex tasks such as pathfinding on a graph. This is in contrast to memory networks, DNCs, and HAM modules, which are restricted to processing only a fixed number of locations in a given timestep. On the other hand, the distributed nature of the GGT-NN model means that it is less time and space efficient than these other networks. Since every node can communicate with every other node, the time and space required to run a GGT-NN step scales quadratically with the size of the input. A DNC or memory network, on the other hand, either scales linearly (since it attends to all stored data or memories) or is constant (if restricted to a fixed-size memory), and a HAM module scales logarithmically (due to the tree structure).

## 7 CONCLUSION

The results presented here show that GGT-NNs are able to successfully model a wide variety of tasks using graph-structured states and potentially could be useful in solving many other types of problems. The specific GGT-NN model described here can be used as-is for tasks consisting of a sequence of input sentences and graphs, optionally followed by a query. In addition, due to the modular nature of GGT-NNs, it is possible to reconfigure the order of the transformations to produce a model suitable for a different task.

The GGT-NN architecture has a few advantages over the architectures described in existing works. In contrast to other approaches to working with structured data, GGT-NNs are designed to work with unstructured input, and are able to modify a graphical structure based on the input. And in contrast to memory networks or DNCs, the internal state of the network is explicitly graph structured, and complex computations can be distributed across the nodes of the graph.

One downside of the current model is that the time and space required to train the model increase very quickly as the complexity of the task increases, which limits the model's applicability. It would be very advantageous to develop optimizations that would allow the model to train faster and with smaller space requirements, such as using sparse edge connections, or only processing some subset of the nodes at each timestep. Another promising direction of future work is in reducing the level of supervision needed to obtain meaningful graphs, for example by combining a few examples that have full graph-level supervision with a larger set of examples that do not have graph-level information, or using additional regularization to enable the GGT-NN model to be trained without any graph information.

There are exciting potential uses for the GGT-NN model. One particularly interesting application would be using GGT-NNs to extract graph-structured information from unstructured textual descriptions. More generally, the graph transformations provided here may allow machine learning to interoperate more flexibly with other data sources and processes with structured inputs and outputs.

### ACKNOWLEDGMENTS

I would like to thank Harvey Mudd College for computing resources. I would also like to thank the developers of the Theano library, which I used to run my experiments. This work used the Extreme Science and Engineering Discovery Environment (XSEDE), which is supported by National Science Foundation grant number ACI-1053575.

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

## APPENDIX A    BACKGROUND ON GG-NNS AND GGS-NNS

This section gives additional background on the implementation of GG-NNs and GGS-NNs, described by Li et al. (2016).

Recall from section 2.2 that GG-NNs represent a graph $\mathcal{G} = (\mathcal{V}, \mathcal{E})$ as a set $V$ of nodes $v$ with unique values $1, \ldots, |\mathcal{V}|$ and a set $\mathcal{E}$ of directed edges $e = (v, v') \in \mathcal{V} \times \mathcal{V}$ oriented from $v$ to $v'$. Each node has an annotation $\mathbf{x}_v \in \mathbb{R}^N$ and a hidden state $\mathbf{h}_v \in \mathbb{R}^D$. Additionally, each edge has a type $y_e \in \{1, \cdots, M\}$.

Initially, $\mathbf{h}_v^{(1)}$ is set to the annotation $\mathbf{x}_v$ padded with zeros. Then nodes exchange information for some fixed number of timesteps $T$ according to the propagation model

$$
\begin{aligned}
\mathbf{h}_v^{(1)} &= [\mathbf{x}_v^\top, \mathbf{0}]^\top & \mathbf{a}_v^{(t)} &= \mathbf{A}_{v:}^\top [\mathbf{h}_1^{(t-1)\top} \cdots \mathbf{h}_{|\mathcal{V}|}^{(t-1)\top}]^\top \\
\mathbf{z}_v^{(t)} &= \sigma(\mathbf{W}_z \mathbf{a}_v^{(t)} + \mathbf{U} \mathbf{h}_v^{(t-1)}) & \mathbf{r}_v^{(t)} &= \sigma(\mathbf{W}_r \mathbf{a}_v^{(t)} + \mathbf{U}_r \mathbf{h}_v^{(t-1)}) \\
\widetilde{\mathbf{h}_v^{(t)}} &= \tanh(\mathbf{W} \mathbf{a}_v^{(t)} + \mathbf{U}(\mathbf{r}_v^{(t)} \odot \mathbf{h}_v^{(t-1)})) & \mathbf{h}_v^{(t)} &= (1 - \mathbf{z}_v^{(t)}) \odot \mathbf{h}_v^{(t-1)} + \mathbf{z}_v^{(t)} \odot \widetilde{\mathbf{h}_v^{(t)}}.
\end{aligned}
\tag{1}
$$

Here $\mathbf{a}_v^{(t)}$ represents the information received by each node from its neighbors in the graph, and the matrix $\mathbf{A} \in \mathbb{R}^{D|\mathcal{V}| \times 2D|\mathcal{V}|}$ has a specific structure that determines how nodes communicate. The first half of $\mathbf{A}$, denoted $\mathbf{A}^{(out)} \in \mathbb{R}^{D|\mathcal{V}| \times D|\mathcal{V}|}$, corresponds to outgoing edges, whereas the second half $\mathbf{A}^{(in)} \in \mathbb{R}^{D|\mathcal{V}| \times D|\mathcal{V}|}$ corresponds to incoming edges.

Each edge type $y$ corresponds to specific forward and backward propagation matrices $\mathbf{P}_y, \mathbf{P}_y' \in \mathbb{R}^{D \times D}$ which determine how to propagate information across an edge of that type in each direction. The $D \times D$-sized submatrix of $\mathbf{A}^{(out)}$ in position $i, j$ contains $\mathbf{P}_y$ if an edge of type $y$ connects nodes $n_i$ to $n_j$, or $\mathbf{0}$ if no such edge connects in that direction. Similarly, the $D \times D$-sized submatrix of the matrix $\mathbf{A}^{(in)}$ in position $i, j$ contains $\mathbf{P}_y'$ if an edge of type $y$ connects nodes $n_j$ to $n_i$, or $\mathbf{0}$ if no such edge connects in that direction. $\mathbf{A}_{v:} \in \mathbb{R}^{D \times 2D|\mathcal{V}|}$ is the submatrix of $\mathbf{A}$ corresponding to node $v$. Thus, multiplication by $\mathbf{A}_{v:}$ in 1 is equivalent to taking the following sum:

$$
\mathbf{a}_v^{(t)} = \sum_{v' \in \mathcal{V}} \left( \sum_{y=1}^{M} s_{\text{edge}}(v, v', y) \odot \mathbf{P}_y + s_{\text{edge}}(v', v, y) \odot \mathbf{P}_y' \right) \mathbf{h}_{v'}^{(t-1)}
\tag{2}
$$

where $s_{\text{edge}}(v, v', y)$ is 1 if $e = (v, v') \in \mathcal{E}$ and $y_e = y$, and 0 otherwise.

The output from a GG-NN is flexible depending on the task. For node selection tasks, a node score $o_v = g(\mathbf{h}_v^{(T)}, \mathbf{x}_v)$ is given for each node, and then a softmax operation is applied. Graph-level outputs are obtained by combining an attention mechanism $i$ and a node representation function $j$, both implemented as neural networks, to produce the output representation

$$
\mathbf{h}_{\mathcal{G}} = \tanh \left( \sum_{v \in \mathcal{V}} \sigma(i(\mathbf{h}_v^{(T)}, \mathbf{x}_v)) \odot \tanh(j(\mathbf{h}_v^{(T)}, \mathbf{x}_v)) \right)
\tag{3}
$$

Gated Graph Sequence Neural Networks (GGS-NN) are an extension of GG-NNs to sequential output $\mathbf{o}^{(1)}, \ldots, \mathbf{o}^{(K)}$. At each output step $k$, the annotation matrix $\mathcal{X}$ is given by $\mathcal{X}^{(k)} = [\mathbf{x}_1^{(k)}, \ldots, \mathbf{x}_{|\mathcal{V}|}^{(k)}]^\top \in \mathbb{R}^{|\mathcal{V}| \times L_\mathcal{V}}$. A GG-NN $\mathcal{F}_\mathbf{o}$ is trained to predict an output sequence $\mathbf{o}^{(k)}$ from $\mathcal{X}^{(k)}$, and another GG-NN $\mathcal{F}_\mathbf{X}$ is trained to predict $\mathcal{X}^{(k+1)}$ from $\mathcal{X}^{(k)}$. Prediction of the output at each step is performed as in a normal GG-NN, and prediction of $\mathcal{X}^{(k+1)}$ from the set of all final hidden states $\mathcal{H}^{(k,T)}$ (after $T$ propagation steps of $\mathcal{F}_\mathbf{X}$) occurs according to the equation

$$
\mathbf{x}_v^{(k+1)} = \sigma \left( j(\mathbf{h}_v^{(k,T)}, \mathbf{x}_v^{(k)}) \right).
$$

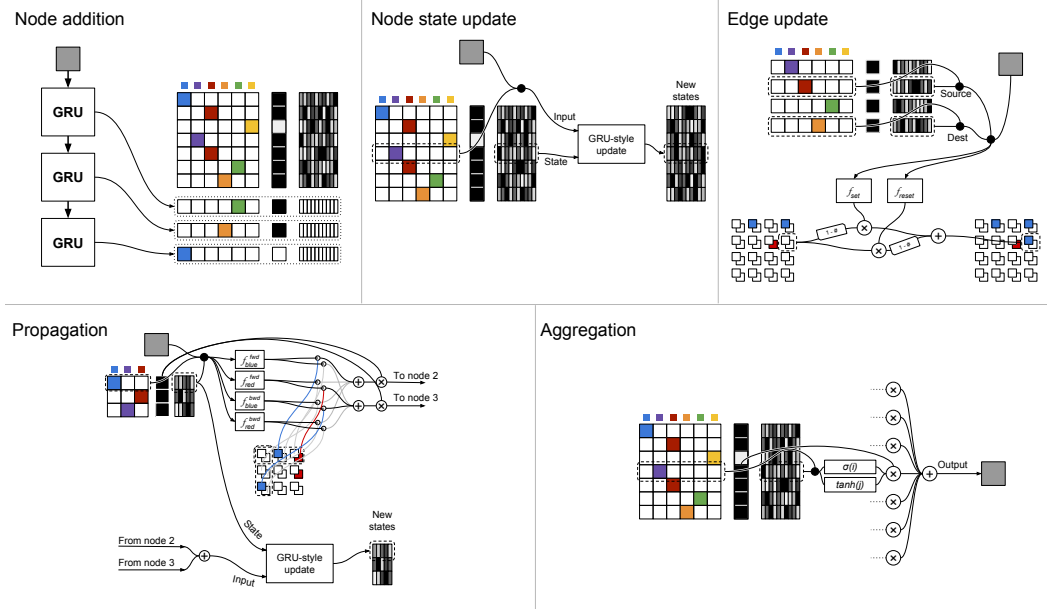

Figure 4: Diagram of the operations performed for each class of transformation. Graph state is shown in the format given by Figure 1. Input and output are shown as gray boxes. Black dots represent concatenation, and $+$ and $\times$ represent addition and multiplication, respectively. $1 - \#$ represents taking the input value and subtracting it from 1. Note that for simplicity, operations are only shown for single nodes or edges, although the operations act on all nodes and edges in parallel. In particular, the propagation section focuses on information sent and received by the first node only. In that section the strengths of the edges in the connectivity matrix determine what information is sent to each of the other nodes. Light gray connections indicate the value zero, corresponding to situations where a given edge is not present.

## APPENDIX B    GRAPH TRANSFORMATION DETAILS

In this section I describe in detail the implementations of each type of differentiable graph transformation.[1] A diagram of the implementation of each transformation is shown in Figure 4. Note that it is natural to think of these transformations as operating on a single graphical state, and each modifying the state in place. However, in the technical descriptions of these transformations, the operations will be described as functions that take in an old graph and produce a new one, similarly to unrolling a recurrent network over time.

### B.1    NODE ADDITION

The node addition transformation $\mathcal{T}_{\text{add}} : \Gamma \times \mathbb{R}^\alpha \to \Gamma$ takes as input a graph $\mathcal{G}$ and an input vector $\mathbf{a} \in \mathbb{R}^\alpha$, and produces a graph $\mathcal{G}'$ with additional nodes. The annotation and strength of each new node is determined by a function $f_{\text{add}} : \mathbb{R}^\alpha \times \mathbb{R}^\beta \to \mathbb{R} \times \mathbb{R}^N \times \mathbb{R}^\beta$, where $\alpha$ is the length of the input vector, $\beta$ is the length of the internal state vector, and as before $N$ is the number of node types. The new nodes are then produced according to

$$\left(s_{|V_{\mathcal{G}}|+i}, \mathbf{x}_{|V_{\mathcal{G}}|+i}, \mathbf{h}_i\right) = f_{\text{add}}(\mathbf{a}, \mathbf{h}_{i-1}), \tag{4}$$

starting with $\mathbf{h}_0$ initialized to some learned initial state, and recurrently computing $s_v$ and $\mathbf{x}_v$ for each new node, up to some maximum number of nodes. Based on initial experiments, I found that implementing $f_{\text{add}}$ as a GRU layer followed by 2 hidden $\tanh$ layers was effective, although other recurrent networks would likely be similarly effective. The node hidden states $\mathbf{h}_v$ are initialized to zero. The recurrence should be computed as many times as the maximum number of nodes that

---

[1]The code for each transformation, and for the GGT-NN model itself, is available at https://github.com/hexahedria/gated-graph-transformer-network.

might be produced. The recurrent function $f_{\text{add}}$ can learn to output $s_v = 0$ for some nodes to create fewer nodes, if necessary.

Note that in order to use information from all of the existing nodes to produce the new nodes, the input to this transformation should include information provided by an aggregation transformation $\mathcal{T}_{\text{repr}}$, described in section B.5.

## B.2 NODE STATE UPDATE

The node state update transformation $\mathcal{T}_{\mathbf{h}} : \Gamma \times \mathbb{R}^{\alpha} \to \Gamma$ takes as input a graph $\mathcal{G}$ and an input vector $\mathbf{a} \in \mathbb{R}^{\alpha}$, and produces a graph $\mathcal{G}'$ with updated node states. This is accomplished by performing a GRU-style update for each node, where the input is a concatenation of $\mathbf{a}$ and that node's annotation vector $\mathbf{x}_v$ and the state is the node's hidden state, according to

$$\mathbf{r}_v = \sigma\left(\mathbf{W}_r[\mathbf{a}\,\mathbf{x}_v] + \mathbf{U}_r\mathbf{h}_v + \mathbf{b}_r\right), \qquad \mathbf{z}_v = \sigma\left(\mathbf{W}_z[\mathbf{a}\,\mathbf{x}_v] + \mathbf{U}_z\mathbf{h}_v + \mathbf{b}_z\right),$$
$$\widetilde{\mathbf{h}}'_v = \tanh\left(\mathbf{W}[\mathbf{a}\,\mathbf{x}_v] + \mathbf{U}\left(\mathbf{r}\odot\mathbf{h}_v\right) + \mathbf{b}\right), \qquad \mathbf{h}'_v = \mathbf{z}_v \odot \mathbf{h}_v + (1 - \mathbf{z}_v) \odot \widetilde{\mathbf{h}}'_v$$

### B.2.1 DIRECT REFERENCE UPDATE

For some tasks, performance can be improved by providing information to nodes of a particular type only. For instance, if the input is a sentence, and one word of that sentence directly refers to a node type (e.g., if nodes of type 1 represent Mary, and Mary appears in the sentence), it can be helpful to allow all nodes of type 1 to perform an update using this information. To accomplish this, $\mathcal{T}_{\mathbf{h}}$ can be modified to take node types into account. (This modification is denoted $\mathcal{T}_{\mathbf{h},\text{direct}}$.) Instead of a single vector $\mathbf{a} \in \mathbb{R}^{\alpha}$, the direct-reference transformation takes in $\mathbf{A} \in \mathbb{R}^{N \times \alpha}$, where $\mathbf{A}_n \in \mathbb{R}^{\alpha}$ is the input vector for nodes with type $n$. The update equations then become

$$\mathbf{a}_v = \mathbf{x}_v \mathbf{A}$$
$$\mathbf{r}_v = \sigma\left(\mathbf{W}_r[\mathbf{a}_v\,\mathbf{x}_v] + \mathbf{U}_r\mathbf{h}_v + \mathbf{b}_r\right), \qquad \mathbf{z}_v = \sigma\left(\mathbf{W}_z[\mathbf{a}_v\,\mathbf{x}_v] + \mathbf{U}_z\mathbf{h}_v + \mathbf{b}_z\right),$$
$$\widetilde{\mathbf{h}}'_v = \tanh\left(\mathbf{W}[\mathbf{a}_v\,\mathbf{x}_v] + \mathbf{U}\left(\mathbf{r}\odot\mathbf{h}_v\right) + \mathbf{b}\right), \quad \mathbf{h}'_v = \mathbf{z}_v \odot \mathbf{h}_v + (1 - \mathbf{z}_v) \odot \widetilde{\mathbf{h}}'_v$$

## B.3 EDGE UPDATE

The edge update transformation $\mathcal{T}_{\mathcal{C}} : \Gamma \times \mathbb{R}^{\alpha} \to \Gamma$ takes a graph $\mathcal{G}$ and an input vector $\mathbf{a} \in \mathbb{R}^{\alpha}$, and produces a graph $\mathcal{G}'$ with updated edges. For each pair of nodes $(v, v')$, the update equations are

$$\mathbf{c}_{v,v'} = f_{\text{set}}(\mathbf{a}, \mathbf{x}_v, \mathbf{h}_v, \mathbf{x}_{v'}, \mathbf{h}_{v'}) \qquad\qquad \mathbf{r}_{v,v'} = f_{\text{reset}}(\mathbf{a}, \mathbf{x}_v, \mathbf{h}_v, \mathbf{x}_{v'}, \mathbf{h}_{v'})$$
$$\mathcal{C}'_{v,v'} = (1 - \mathcal{C}_{v,v'}) \odot \mathbf{c}_{v,v'} + \mathcal{C}_{v,v'} \odot (1 - \mathbf{r}_{v,v'}).$$

The functions $f_{\text{set}}, f_{\text{reset}} : \mathbb{R}^{\alpha \times 2N \times 2D} \to [0,1]^Y$ are implemented as neural networks. (In my experiments, I used a simple 2-layer fully connected network.) $\mathbf{c}_{v,v',y}$ gives the level of belief in $[0,1]$ that an edge from $v$ to $v'$ of type $y$ should be created if it does not exist, and $\mathbf{r}_{v,v',y}$ gives the level of belief in $[0,1]$ that an edge from $v$ to $v'$ of type $y$ should be removed if it does. Setting both to zero results in no change for that edge, and setting both to 1 toggles the edge state.

## B.4 PROPAGATION

The propagation transformation $\mathcal{T}_{\text{prop}} : \Gamma \to \Gamma$ takes a graph $\mathcal{G} = \mathcal{G}^{(0)}$ and runs a series of $T$ propagation steps (as in GG-NN), returning the resulting graph $\mathcal{G}' = \mathcal{G}^{(T)}$. The GG-NN propagation step is extended to handle node and edge strengths, as well as to allow more processing to occur to

the information transferred across edges. The full propagation equations for step $t$ are

$$\mathbf{a}_v^{(t)} = \sum_{v' \in \mathcal{V}} s_{v'} \sum_{y=1}^{M} \mathcal{C}_{v,v',y} \odot f_y^{\text{fwd}}(\mathbf{x}_{v'}, \mathbf{h}_{v'}^{(t-1)}) + \mathcal{C}_{v',v,y} \odot f_y^{\text{bwd}}(\mathbf{x}_{v'}, \mathbf{h}_{v'}^{(t-1)}) \tag{5}$$

$$\mathbf{z}_v^{(t)} = \sigma(\mathbf{W}_z[\mathbf{a}_v^{(t)} \ \mathbf{x}_v] + \mathbf{U}\mathbf{h}_v^{(t-1)} + \mathbf{b}_z) \tag{6}$$

$$\mathbf{r}_v^{(t)} = \sigma(\mathbf{W}_r[\mathbf{a}_v^{(t)} \ \mathbf{x}_v] + \mathbf{U}_r\mathbf{h}_v^{(t-1)} + \mathbf{b}_r) \tag{7}$$

$$\widetilde{\mathbf{h}_v^{(t)}} = \tanh(\mathbf{W}[\mathbf{a}_v^{(t)} \ \mathbf{x}_v] + \mathbf{U}(\mathbf{r}_v^{(t)} \odot \mathbf{h}_v^{(t-1)}) + \mathbf{b}_h) \tag{8}$$

$$\mathbf{h}_v^{(t)} = (1 - \mathbf{z}_v^{(t)}) \odot \mathbf{h}_v^{(t-1)} + \mathbf{z}_v^{(t)} \odot \widetilde{\mathbf{h}_v^{(t)}}. \tag{9}$$

Equation 5 has been adjusted in the most significant manner (relative to Equation 2). In particular, $s_{v'}$ restricts propagation so that nodes with low strength send less information to adjacent nodes, $s_{\text{edge}}$ has been replaced with $\mathcal{C}$ to allow edges with fractional strength, and the propagation matrices $\mathbf{P}_y, \mathbf{P}'_y$ have been replaced with arbitrary functions $f_y^{\text{fwd}}, f_y^{\text{bwd}} : \mathbb{R}^N \times \mathbb{R}^D \to \mathbb{R}^\alpha$, where $\alpha$ is the length of the vector $\mathbf{a}$. I used a fully connected layer to implement each function in my experiments. Equations 6, 7, and 8 have also been modified slightly to add a bias term.

## B.5 AGGREGATION

The aggregation transformation $\mathcal{T}_{\text{repr}} : \Gamma \to \mathbb{R}^\alpha$ produces a graph-level representation vector from a graph. It functions very similarly to the output representation of a GG-NN (equation 3), combining an attention mechanism with a node representation function, but is modified slightly to take into account node strengths. As in GG-NN, both $i$ and $j$ are neural networks, and in practice a single fully connected layer appears to be adequate for both.

$$\mathbf{h}_\mathcal{G} = \tanh\left(\sum_{v \in \mathcal{V}} s_v \sigma(i(\mathbf{h}_v^{(T)}, \mathbf{x}_v)) \odot \tanh(j(\mathbf{h}_v^{(T)}, \mathbf{x}_v))\right).$$

## APPENDIX C  GRAPH FORMAT DETAILS

### C.1  bAbI TASKS

The knowledge graph object used during generation of the bAbI tasks is structured as a dictionary relating entities to each other with specific relationship types. Entities are identified based on their names, and include people (John, Mary, Sandra), locations (bedroom, kitchen, garden), objects (football, apple, suitcase), animals (mouse, wolf, cat), and colors (white, yellow, green), depending on the particular task. Relationships between entities are also expressed as strings, and are directed: if John is holding the milk there is an "is_in" relationship from "milk" to "John"; if Sandra is in the bedroom there is an "is_in" relationship from "Sandra" to "bedroom"; if Lily is green there is a "has_color" relationship from "Lily" to "green", etc.

The transformation from the knowledge object to a graph is straightforward: each entity used is assigned to a new node type, and relationships between entities are represented as edges between the corresponding nodes. To avoid confusion from overloaded relationships (such as "is_in" being used to represent an object being held by a person as well as a person being in a room), relation names are given a distinct edge type depending on the usage context. For instance, when a person is carrying an object, the generic "is_in" relationship becomes an edge of type "gettable_is_in_actor".

Some of the graph representations had to be modified in order to ensure that they contained all of the necessary information. For instance, task 3 requires the network to remember where items were in the past, but the knowledge object only contained references to their current locations. In these cases, a linked list structure was added to the knowledge object to allow the history information to be represented in the graph.

In particular, each time an item changed locations, a new "record" node was added, with a "previous" edge to the previous history node and a "value" edge to the current location of the item. Each item then connected to the most recent history node using a "history-head" edge. This ensures that the history of each node is present in the graph.

1. John grabbed the milk.
2. John travelled to the bedroom.
3. Sandra took the football.
4. John went to the garden.
5. *John let go of the milk.*
6. Sandra let go of the football.
7. John got the football.
8. John grabbed the milk.
Where is the milk?

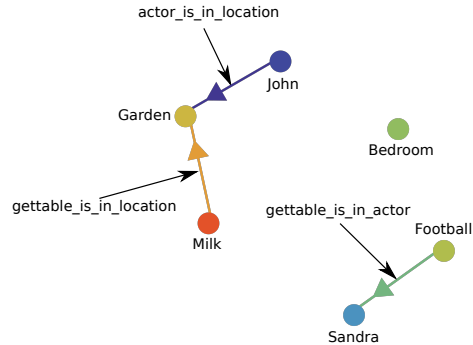

Figure 5: Diagram of one sample story from the bAbI dataset (Task 2), along with a graphical representation of the knowledge state after the italicized sentence.

| | |
|---|---|
| 1. init 1 | 11. init 1 |
| 2. init 1 | 12. init 1 |
| 3. init 1 | 13. init 0 |
| 4. init 1 | 14. simulate |
| 5. init 1 | 15. simulate |
| 6. init 0 | 16. simulate |
| 7. init 0 | 17. *simulate* |
| 8. init 0 | 18. simulate |
| 9. init 1 | 19. simulate |
| 10. init 1 | 20. simulate |

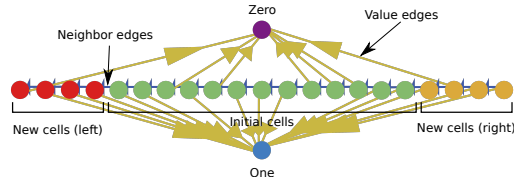

Figure 6: Diagram of one example from the automaton task, along with a graphical representation of the automaton state after the fourth simulate command (italicized).

| | |
|---|---|
| 1. rule state_3 0 0 state_0 L | 10. input symbol_0 head |
| 2. rule state_1 0 1 state_0 R | 11. input symbol_0 |
| 3. rule state_2 1 1 state_2 L | 12. input symbol_0 |
| 4. rule state_3 1 0 state_3 L | 13. input symbol_1 |
| 5. rule state_0 1 0 state_0 R | 14. run |
| 6. rule state_0 0 1 state_2 N | 15. *run* |
| 7. rule state_2 0 0 state_2 R | 16. run |
| 8. rule state_1 1 1 state_0 N | 17. run |
| 9. start state_1 | 18. run |
| | 19. run |

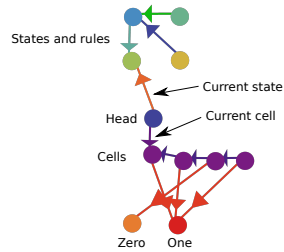

Figure 7: Diagram of an example from the Turing machine task, with a graphical representation of the machine state after the second run command (italicized).

In a few of the tasks, specific entities had multi-word representations. While this works for normal input, it makes it difficult to do direct reference, since direct reference is checked on an individual word level. These tasks were modified slightly so that the entities are referred to with single words (e.g. "red_square" instead of "red square").

An example of a graph produced from the bAbI tasks is given in Figure 5.

## C.2 CELLULAR AUTOMATON

The cellular automaton task was mapped to graphical format as follows: Nodes have 5 types: zero, one, init-cell, left-cell, and right-cell. Edges have 2 types: value, and next-r. There is always exactly one "zero" node and one "one" node, and all of the cell nodes form a linked list, with a "value" edge connecting to either zero or one, and a "next-r" edge pointing to the next cell to the right (or no edge for the rightmost cell).

At the start of each training example, there are 13 timesteps with input of the form "init X" where X is 0 or 1. These timesteps indicate the first 13 initial cells. Afterward, there are 7 "simulate" inputs. At each of these timesteps, one new left-cell node is added on the left, one new right-cell node is added on the right, and then all cells update their value according to the Rule 30 update rules.

An example of the graphical format for the cellular automaton task is given in Figure 6.

## C.3 TURING MACHINE

For the Turing machine task, nodes were assigned to 8 types: state-A, state-B, state-C, state-D, head, cell, 0, and 1. Edges have 16 types: head-cell, next-left, head-state, value, and 12 types of the form rule-R-W-D, where R is the symbol read (0 or 1), W is the symbol written (0 or 1), and D is the direction to move afterward (Left, Right, or None). State nodes are connected with rule edges, which together specify the rules governing the Turing machine. Cell nodes are connected to adjacent cells with next-left edges, and to the symbol on the tape with value edges. Finally, the head node is connected to the current state with a head-state edge, and to the current cell of the head with a head-cell edge.

At the start of each training example, each of the rules for the Turing machine are given, in the form "rule state-X R W state-Y D". Next, the initial state is given in the format "start state-X", and the initial contents of the tape (of length 4) are given sequentially in the format "input symbol-X", with the position for the head to start marked by "input symbol-X head". Finally, there are 6 "run" inputs, after each of which the head node updates its edges and the cell at the head updates its value according to the rules of the Turing machine. If the head leaves the left or right of the tape, a new node is introduced there.

An example of the graphical format for the Turing machine task is given in Figure 7.

## APPENDIX D   GRAPH SEQUENCE INPUT

The model described in Section 4 conditions the output of the model on the final graph produced by the network. This is ideal when the graph represents all of the necessary knowledge for solving the task. However, it may also be desirable for each graph to represent a subset of knowledge corresponding to a particular time, and for the output to be based on the sequence of graphs produced. For instance, in the third bAbI task (which requires reasoning about the temporal sequence of events) each graph could represent the state of the word at that particular time, instead of representing the full sequence of events prior to that time. In Appendix C, section C.1, I describe a transformation to the tasks which allows all information to be contained in the graph. But this adds complexity to the graphical structure. If it were possible for the model to take into account the full sequence of graphs, instead of just the final one, we could maintain the simplicity of the graph transformation.

To this end, I present an extension of the GGT-NN model that can produce output using the full graphical sequence. In the extended model, the graphical output of the network after each input sentence is saved for later use. Then, when processing the query, the same set of query transformations are applied to every intermediate graph, producing a sequence of representation vectors $h_1^{answer}, \ldots, h_K^{answer}$. These are then combined into a final summary representation vector $h_{summary}^{answer}$

| Task | Direct reference Accuracy | No direct reference Accuracy |
|------|---------------------------|------------------------------|
| 3 - Three Supporting Facts | 90.3% | 65.4% |
| 5 - Three Arg. Relations | 89.8% | 74.2% |

Table 4: Performance of the sequence-extended GGT-NN on the two bAbI tasks with a temporal component.

---

**Algorithm 2** Sequence-Extended Pseudocode

$\mathcal{G}_0 \leftarrow \varnothing$ $\triangleright$ Initialize $\mathcal{G}$ to an empty graph
**for** $k$ from 1 to $K$ **do** $\triangleright$ Process each sentence
 $\mathcal{G}_k \leftarrow \mathcal{T}_{\mathbf{h}}(\mathcal{G}_{k-1}, \mathbf{i}^{(k)})$
 **if** direct reference enabled **then**
 $\mathcal{G}_k \leftarrow \mathcal{T}_{\mathbf{h}}^{\text{direct}}(\mathcal{G}_k, \mathbf{D}^{(k)})$
 **end if**
 **if** intermediate propagation enabled **then**
 $\mathcal{G}_k \leftarrow \mathcal{T}_{\text{prop}}(\mathcal{G}_k)$
 **end if**
 $\mathbf{h}_{\mathcal{G}_k}^{\text{add}} \leftarrow \mathcal{T}_{\text{repr}}(\mathcal{G}_k)$
 $\mathcal{G}_k \leftarrow \mathcal{T}_{\text{add}}(\mathcal{G}_k, [\mathbf{i}^{(k)} \ \mathbf{h}_{\mathcal{G}_k}^{\text{add}}])$
 $\mathcal{G}_k \leftarrow \mathcal{T}_{\mathcal{C}}(\mathcal{G}_k, \mathbf{i}^{(k)})$
**end for**
$\mathbf{h}_{\text{summary}}^{\text{answer}} \leftarrow \mathbf{0}$ $\triangleright$ Initialize $\mathbf{h}_{\text{summary}}^{\text{answer}}$ to the zero vector
**for** $k$ from 1 to $K$ **do** $\triangleright$ Process the query for each graph
 $\mathcal{G}_k \leftarrow \mathcal{T}_{\mathbf{h}}^{\text{query}}(\mathcal{G}_k, \mathbf{i}^{\text{query}})$
 **if** direct reference enabled **then**
 $\mathcal{G}_k \leftarrow \mathcal{T}_{\mathbf{h}}^{\text{query,direct}}(\mathcal{G}_k, \mathbf{D}^{\text{query}})$
 **end if**
 $\mathcal{G}_k \leftarrow \mathcal{T}_{\text{prop}}^{\text{query}}(\mathcal{G}_k)$
 $\mathbf{h}_{\mathcal{G}_k}^{\text{answer}} \leftarrow \mathcal{T}_{\text{repr}}^{\text{query}}(\mathcal{G}_k)$
 $\mathbf{h}_{\text{summary}}^{\text{answer}} \leftarrow f_{\text{summarize}}(\mathbf{h}_{\mathcal{G}_k}^{\text{answer}}, \mathbf{h}_{\text{summary}}^{\text{answer}})$
**end for**
**return** $f_{\text{output}}(\mathbf{h}_{\text{summary}}^{\text{answer}})$

---

using a recurrent network such as a GRU layer, from which the output can be produced. The modified pseudocode for this is shown in Algorithm 2.

I evaluated the extended model on bAbI tasks 3 and 5, the two tasks which asked questions about a sequence of events. (Note that although Task 14 also involves a sequence of events, it uses a set of discrete named time periods and so is not applicable to this modification.) The model was trained on each of these tasks, without the extra record and history nodes used to store the sequence, instead simply using the sequence of graphs to encode the relevant information. Due to the simpler graphs produced, intermediate propagation was also disabled.

Results from training the model are shown in Table 4. The accuracy of the extended model appears to be slightly inferior to the original model in general, although the extended direct-reference model of task 5 performs slightly better than its original counterpart. One possible explanation for the inferiority of the extended model is that the increased amount of query processing made the model more likely to overfit on the training data. Even so, the extended model shows promise, and could be advantageous for modeling complex tasks for which preprocessing the graph would be impractical.

