# Peer review of "Learning Graphical State Transitions"

_ICLR 2017 — accepted_

[Public Comment · Daniel Tarlow · 01 Dec 2016]
**Cool paper!**

Great job with the paper. I love the idea.

It seems the main weakness is the lack of broader context. Specifically,

Baselines: 
- it'd be nice to see bAbI results from other papers in Table 1, to quickly get a sense for how your results compare
- have you thought about baseline methods for the experiments in 4.2?
- are there any other tasks like bAbI where you can compare GGT-NN to existing results from the literature?

Related work:
- I'd appreciate a related work section with discussion of the similarities and differences to other models with more or less structured memory representations including, e.g., Memory Networks, Hierarchical Attentive Memory, and Differentiable Neural Computers (DNC).

Finally, have you considered training your model with a mixture of strong and weak supervision? Maybe you could get away with just a few instances labeled with the strong supervision described in 3.1, with the rest labeled only by the correct answer?

[Author Response · Daniel D. Johnson · 13 Dec 2016]
**Revision uploaded**

Based on the reviewer comments, I have uploaded a new revision of the paper, with the following changes:

- Added figures depicting the differentiable graph format and each of the graph transformations
- Added a section comparing the GGT-NN model to existing works
- Included baselines for the bAbI tasks, with discussion of the relative performance of the GGT-NN model
- Clarified the feasibility of alternative network configurations (relative to Algorithm 1)
- Moved implementation details of the graph transformations into the appendix
- Clearly separated and simplified the background information section
- Clarified the training procedure for the GGT-NN model

Thank you very much for your feedback.

[Official Review · AnonReviewer2 · rating 9 · confidence 3 · 16 Dec 2016]
**No Title**

The paper proposes an extension of the Gated Graph Sequence Neural Network by including in this model the ability to produce complex graph transformations. The underlying idea is to propose a method that will be able build/modify a graph-structure as an internal representation for solving a problem, and particularly for solving question-answering problems in this paper. The author proposes 5 different possible differentiable transformations that will be learned on a training set, typically in a supervised fashion where the state of the graph is given at each timestep. A particular occurence of the model is presented that takes a sequence as an input a iteratively update an internal graph state to a final prediction, and which can be applied for solving QA tasks (e.g BaBi) with interesting results.

The approach  in this paper is really interesting since the proposed model is able to maintain a representation of its current state as a complex graph, but still keeping the property of being differentiable and thus easily learnable through gradient-descent techniques. It can be seen as a succesfull attempt to mix continuous and symbolic representations. It moreover seems more general that the recent attempts made to add some 'symbolic' stuffs in differentiable models (Memory networks, NTM, etc...) since the shape of the state is not fixed here and can evolve. My main concerns is about the way the model is trained i.e by providing the state of the graph at each timestep which can be done for particular tasks (e.g Babi) only, and cannot be the solution for more complex problems. My other concern is about the whole content of the paper that would perhaps best fit a journal format and not a conference format, making the article still difficult to read due to its density.

[Official Review · AnonReviewer1 · rating 9 · confidence 3 · 19 Dec 2016 (modified: 20 Jan 2017)]
**Complex implementation of a differentiable memory as a graph with promising preliminary results.**

This paper proposes learning on the fly to represent a dialog as a graph (which acts as the memory), and is first demonstrated on the bAbI tasks. Graph learning is part of the inference process, though there is long term representation learning to learn graph transformation parameters and the encoding of sentences as input to the graph. This seems to be the first implementation of a differentiable memory as graph: it is much more complex than previous approaches like memory networks without significant gain in performance in bAbI tasks, but it is still very preliminary work, and the representation of memory as a graph seems much more powerful than a stack. Clarity is a major issue, but from an initial version that was constructive and better read by a computer than a human, the author proposed a hugely improved later version. This original, technically accurate (within what I understood) and thought provoking paper is worth publishing.

The preliminary results do not tell us yet if the highly complex graph-based differentiable memory has more learning or generalization capacity than other approaches. The performance on the bAbI task is comparable to the best memory networks, but still worse than more traditional rule induction (see

[Official Review · AnonReviewer3 · rating 7 · confidence 2 · 20 Dec 2016]
**Architecture which allows to learn graph->graph tasks,  improves state of the art on babi**

The main contribution of this paper seems to be an introduction of a set of differential graph transformations which will allow you to learn graph->graph classification tasks using gradient descent. This maps naturally to a task of learning a cellular automaton represented as sequence of graphs. In that task, the graph of nodes grows at each iteration, with nodes pointing to neighbors and special nodes 0/1 representing the values. Proposed architecture allows one to learn this sequence of graphs, although in the experiment, this task (Rule 30) was far from solved.

This idea is combined with ideas from previous papers (GGS-NN) to allow the model to produce textual output rather than graph output, and use graphs as intermediate representation, which allows it to beat state of the art on BaBi tasks.

[Author Response · Daniel D. Johnson · 21 Dec 2016]
**Revision uploaded**

The paper has been updated with the following changes:

- Fixed typo in the equations given in B.2 and B.2.1.
- Added a reference to work done by Giles et al. in Section 6.
- Clarified the operation of direct reference in Sections 3 and 4.
- Added a link to the source code for the model (in Appendix B).
- Minor wording changes in the Abstract, Introduction, and Sections 3 and 4.

[Public Comment · David Liu · 27 Dec 2016]
**Loss function requires strong supervision**

Section 4.1 Supervision mentioned that the proposed method requires additional supervision by providing the correct graph at each timestep. 

I have some questions about whether this is practical: 

1. In some applications, human may or may not be able to provide the "correct graph" at each timestep. What if the human supervisor can only provide suboptimal graphs during training? How would that affect the results?

2. What was the result of not providing such intermediate supervision? Was the result slightly worse than the current result, or significantly different?

3. Such additional supervision would also require more "work" by the human. How can the amount of work be quantified, so the reader understands the implications of having to provide additional supervision?

4. I would appreciate if the need of such strong supervision was mentioned earlier in the paper, perhaps during the introduction or literature review, to give the reader some "warning". Some literature review/comparison of other approaches that require strong supervision would also be appreciated.

Thanks for the good work, I enjoyed reading it!

[Final Decision · Program Chairs · 06 Feb 2017]
**ICLR committee final decision**

The idea of building a graph-based differentiable memory is very good. The proposed approach is quite complex, but it is likely to lead to future developments and extensions. The paper has been much improved since the original submission. The results could be strengthened, with more comparisons to existing results on bAbI and baselines on the experiments here. Exploring how it performs with less supervision, and different types of supervision, from entirely labeled graphs versus just node labels, would be valuable.